# Has the Establishment of High-Tech Zones Improved Urban Economic Resilience? Evidence from Prefecture-Level Cities in China

**Ruoxi Yu [1], Xingneng Xia [1], Tao Huang [1], Sheng Zhang [1,*] and Wenguang Zhou [2]**

[1] School of Public Policy and Administration, Xi'an Jiaotong University, Xi'an 710049, China; yuruoxi@stu.xjtu.edu.cn (R.Y.); xiaxingneng1991@stu.xjtu.edu.cn (X.X.); htao1996@stu.xjtu.edu.cn (T.H.)
[2] School of Public Management, Northwest University, Xi'an 710127, China; zhouwenguang0411@126.com
* Correspondence: zhangsheng_xjtu@163.com

**Abstract:** The establishment of high-tech zones in China represents a significant policy tool aimed at fostering urban scientific and technological innovation while ensuring steady and sustainable economic growth. Using high-tech zones as a quasi-natural experiment and 233 prefecture-level cities in China from 1990 to 2021 as a research sample, this article constructs a difference-in-difference model to test the impact of high-tech zones on urban economic resilience. Our findings reveal several key insights. First, high-tech zones play a crucial role in enhancing urban economic resilience, which is robust across multiple tests. Second, there is significant variation in the influence of high-tech zones on urban economic resilience. Large cities, cities in the Yangtze River Economic Belt (YEB), and eastern cities are more affected than other cities. Third, improving urban innovation ability and optimizing resource allocation are important ways through which high-tech zones influence urban economic resilience. These findings contribute significantly to the evaluation of the high-tech zones policy and form empirical evidence of the policy arrangements' regional-level impact on economic resilience.

**Keywords:** high-tech zones; urban economic resilience; difference-in-differences method

## 1. Introduction

The global economy is currently facing various economic upheavals and challenges that are significantly impacting diverse nations and regions. There are also lingering effects from the COVID-19 outbreak [1,2]. Global trade frictions have greatly hindered economic growth [3–5]. Climate change and sustainability concerns continue to pose threats to the global economy [6,7]. While technological advancements offer numerous business opportunities, they also bring a set of challenges [8–10]. These interconnected disruptions and challenges create a complex and diverse economic landscape, which requires governments to implement appropriate policies and measures to foster sustainable economic growth and development.

Economic resilience is crucial in today's global economic environment, which is characterized by turbulence and challenges [11–13]. The concept of economic resilience comes from other disciplines. Initially, resilience was viewed primarily through the lens of equilibrium theory, which focuses on the speed and extent to which systems return to their original state and trajectory, as seen in engineering resilience and ecological resilience [14–16]. As the concept of resilience was further explored, it incorporated adaptive theory and adopted an evolutionary perspective [17,18]. Emphasis was placed on the system's ability to adapt to unexpected shocks and it was argued that resilience is a continual process [17,19]. In this context, resilience was introduced into spatial economics by Reggiani et al. [20], who defined it as a socioeconomic system's capacity to bounce back from shocks. Expanding on this, Simmie and Martin [17] provided a more comprehensive explanation of economic

resilience, defining it as the capacity of a country's or region's economic system to adjust to shocks and uncertainties from the outside world. Strong economic resilience can mitigate the effects of external shocks and expedite the economy's adaptation and recuperation [17,21–23].

Urban economic resilience, according to regional economic resilience, is the capacity of a city's economic systems to withstand shocks, bounce back quickly from calamities, and improve their capacity for adaptation. Economic resilience encompasses several important dimensions. First, vulnerability is defined as a city's susceptibility to various shocks. Second, resistance signifies a city's ability to withstand the impact of shocks. Third, reorientation describes the capacity of a city's economy to adjust and maintain stable employment and growth rates during times of crisis. Lastly, recoverability refers to a city's economic system's capacity to adjust to sustain steady rates of employment and growth during a crisis [22,24–27]. High economic resilience plays a crucial role in helping regions effectively navigate uncertain events, ensuring the sustainability of the environment, society, and the overall economy. As a result, policymakers, academics, businesses, and communities have increasingly focused their research and attention on economic resilience to ensure regional prosperity and stability [21,24,25,28,29].

In the progression of society and the economy, institutions play a pivotal role [30–32]. China's remarkable economic progress over the past forty years can be attributed to positive changes and institutional innovation, such as the high-tech zones policy [33]. This policy has been instrumental in supporting scientific and technological innovation in urban areas, as well as fostering quality economic development. It has achieved this by establishing spatial platforms that facilitate the concentration of innovation factors [34,35]. By directing capital, talent, and resources to high-tech, high-value-added areas, these zones stimulate regional economic growth, foster agglomeration effects, and promote the formation of industrial clusters to achieve economies of scale [36]. National high-tech zones are currently entering the "third venture" stage, highlighting their prominent position in the area. In this context, high-tech zones have two roles to play: they must support development and growth while also assisting the local economy in adapting to long-term shifts in the economic environment and fostering sustainable growth. This paper aims to address three key questions: Firstly, can urban economic resilience be enhanced by the establishment of high-tech zones? Secondly, is the influence of high-tech zones on urban economic resilience heterogeneous? Lastly, how do high-tech zones affect urban economic resilience; what is the underlying mechanism behind this? For China and other nations, the answers to these questions will be crucial because they will help make urban areas more resilient to both internal and external disturbances, while fostering steady and sustainable economic growth.

The existing literature on this topic primarily focuses on two main areas: the assessment of economic resilience and the study of the factors that influence it. In terms of the assessment of economic resilience, there are two methods commonly used. One approach involves evaluating a region's economic resilience by developing a comprehensive set of indicators [37,38]. However, there is a lack of consensus on the selection of indicators, leading to significant variations in their formulation and raising concerns about the accuracy of measurement outcomes. The second approach involves constructing the economic resilience index by choosing a central variable that represents the change in response to an unexpected disturbance [27]. Most studies choose regional employment or unemployment data, as well as GDP data, as the core variable for this purpose [39,40].

The second major area of research focuses on the factors that influence regional economic resilience. Economic resilience is a complex system influenced by multiple factors [22]. For instance, researchers have analyzed how regional economic foundations and structures, including the economic level and industrial structure, influence economic resilience [27,41–45]. Other scholars have focused on the influence of resource factors, such as infrastructure, human capital, and research institutions, on economic resilience [46,47]. Moreover, several researchers have investigated the influence of institutional environmental factors, specifically policy arrangements, on economic resilience [48–50]. Furthermore, new

perspectives in economic resilience research, such as technological network structures [51] and industry embeddedness [45], have emerged, providing a deeper understanding of this topic. In conclusion, economic resilience is a multifaceted and regionally heterogeneous phenomenon influenced by a combination of factors and their interactions. The high-tech zones policy is widely acknowledged as a successful institutional innovation due to its positive impact on economic development and scientific and technological innovation. Consequently, it is crucial to evaluate whether it has truly contributed to urban economic resilience.

This paper examines the influence of high-tech zones on urban economic resilience. This study makes three significant contributions to the field. Firstly, it presents robust results by analyzing data from a significant sample size of 233 prefecture-level cities in China over a period of 32 years (1990–2021). This enables a more scientific and systematic assessment of the direct impact of high-tech zones on urban economic resilience. Secondly, it compares prefecture-level cities that have established national high-tech zones with those that have not, providing a comprehensive assessment of the policy effects of high-tech zones on urban economic resilience. Thirdly, this study reveals the specific theoretical mechanism connecting high-tech zones and urban economic resilience. This enhances our comprehension of how the policy arrangement of high-tech zones contributes to economic stability and long-term development. Additionally, it offers fresh evidence for sustainable development dimensions that have not been thoroughly investigated.

## 2. Policy Background and Mechanism Analysis

### 2.1. Policy Background

Development zones are policy tools used by various countries to promote economic development in specific regions. These zones have several main objectives, such as attracting foreign direct investment, promoting concentrated industrial growth, encouraging technological innovation, creating employment opportunities, and accelerating regional economic progress [52–56]. Consequently, different countries have established their own development zones, including free trade zones, industrial parks, economic and technological development zones, and industrial or science and technology parks [53,57–59].

China has been actively implementing policies of reform and opening up since the 1980s. This policy includes opening up to the international market, attracting direct foreign investment, and bringing in cutting-edge technology and managerial know-how. The year 1985 witnessed the issuance of the Decision on the Reform of the Science and Technology System by the Central Committee of the Communist Party of China. The goal of this crucial policy is to speed up and expand the implementation of scientific and technological achievements in production, efficiently harnessing the expertise of scientific and technological professionals, and ultimately achieving the liberation of scientific and technological productivity for social progress. Recognizing the significance of innovation and scientific and technological development in driving economic growth, the Chinese government undertook a reformation of their science and technology framework. In 1988, the Chinese Ministry of Science and Technology initiated the 'Torch Program' with the aim of expediting China's science, technology, and economic modernization endeavors. The objective of this program is to strengthen China's standing in the worldwide science, technology, and innovation domains by facilitating and backing high-tech sectors. One of the key components of the Torch Program entails establishing a sequence of high-tech zones.

The Zhongguancun high-tech zone, located in Beijing, was established by China in 1998 as its inaugural high-tech zone. It holds a prominent position as the forerunner among the high-tech zones in China and has made significant contributions to the advancement of technological innovation and the promotion of entrepreneurship. Numerous cities have established their individual high-tech industrial development zones, replicating this successful model. These zones serve as catalysts for innovation in science and technology, upgrading industries and promoting economic development. They are the driving factors behind the expansion of high-tech industries in China. In 1988, China established its initial

high-tech zone, marking the beginning of a 36-year journey. As of the end of 2021, the number of national high-tech zones has increased significantly to 169. Figure 1 illustrates the distribution of cities that host national high-tech zones in 2021.

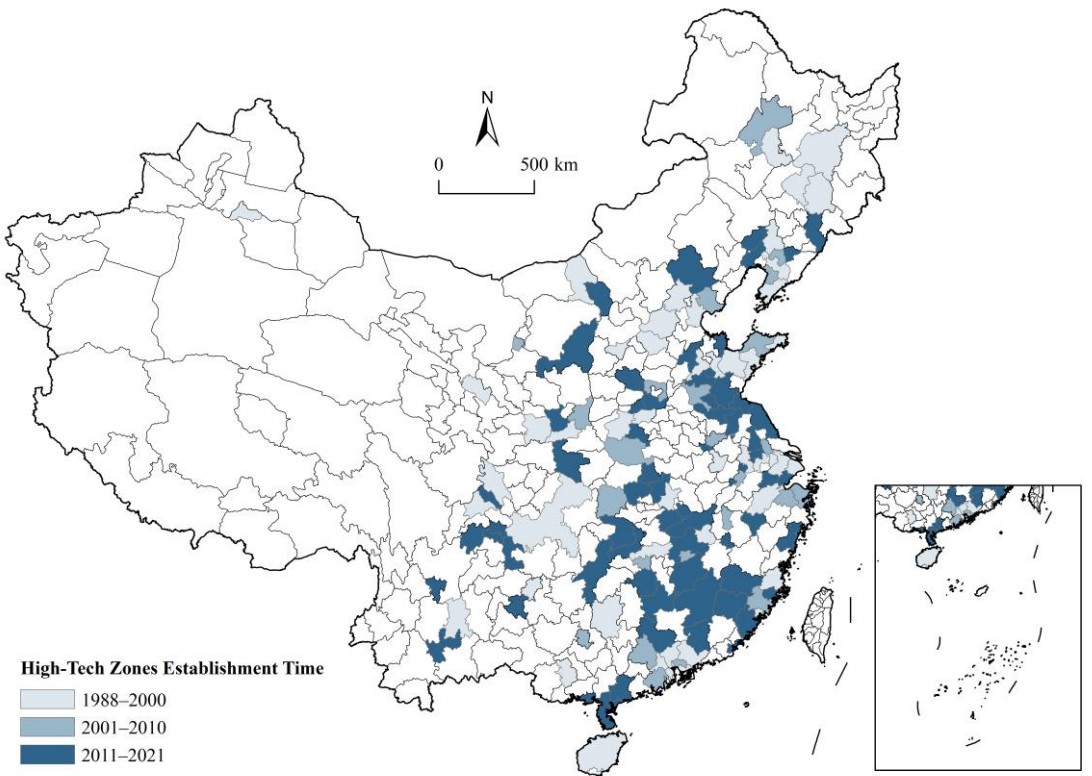

**Figure 1.** Spatial distribution of cities that host national high-tech zones in 2021.

The development of high-tech zones can be categorized into three distinct stages. The initial stage (1988–2000) focused on the 'first venture' and prioritized 'industrialization'. The main objective is to attract investment by leveraging various factors and overcoming the challenges posed by a weak industrial base. The second stage (2001–2008) shifted the development goal to 'developing science and technology industrial parks' from 'building industrial parks'. The objective was to gather and foster innovative resources, nurturing independent innovation capabilities. The third stage (2009 to present) is characterized by the theme of a 'third venture'. High-tech zones have expanded their position from 'science and technology development zones' to hubs with comprehensive innovation factors. They currently function as exemplary zones for the advancement of innovation-driven growth and are a forerunner in the realm of development characterized by excellent quality. Their new mission is to transform into a leading hub for high-tech industries, embrace the knowledge economy, cultivate a culture of innovation, promote modern ecological civilization, and build a harmonious community.

### 2.2. Mechanism Analysis

High-tech zones are innovation ecosystems that bring together individuals and organizations from various areas, including research institutes, universities, enterprises, and innovative entrepreneurs. The primary objective of these zones is to foster science, technology, and innovation, while also promoting economic development and creative activities [60–62]. To bolster the regional economy, they actively endorse the advancement of talent, industries, and innovation chains, thereby fostering urban resilience and driving stable and sustainable growth. Thus, the establishment of high-tech zones is closely intertwined with the enhancement of urban economic resilience.

First, the establishment of high-tech zones is in line with the goal of strengthening urban economic resilience. High-tech zones mainly aim at attaining self-sufficiency and the advancement of science and technology. They actively contribute to the advancement of talent, industry, and innovation integration, which greatly contributes to the transition of development patterns and facilitates high-level growth [63]. This, in turn, contributes to stable and sustainable economic growth, supporting the overall well-being and positive progress of the national economy. Moreover, it is crucial to strengthen the economic resilience of cities to ensure continued urban growth and to enable cities to adapt effectively, withstand shocks, and face challenges from both internal and external factors. By doing so, cities can ensure their long-term prosperity and sustainable development, avoiding severe economic recessions or crises [13,22]. Therefore, both objectives ultimately aim to foster stable and sustainable economic growth.

Second, there is a common concept shared between the establishment of high-tech zones and the enhancement of urban economic resilience. In line with the current trend of sustainable development, high-tech zones actively implement this new development concept. They explore green, low-carbon, and sustainable development paths to attain the goals of carbon peak and carbon neutrality. Additionally, high-tech zones strive to lead in the development of new energy, new technology, and novel business models within green and low-carbon industries [52,55,64,65]. The concept of urban economic resilience emphasizes elements such as economic diversity, scientific and technological innovation, resource sustainability, crisis response, and international competitiveness. Its primary goal is to foster environmentally friendly growth and enhance the economy's ability to adapt to future development [66,67]. Therefore, the common concepts shared by both high-tech zones and urban economic resilience are low-carbon, green, inclusivity, and sustainability.

High-tech zones are a crucial component of a city's economic resilience. Aligned with the national strategy, these zones aim to attract high-tech industries, foster innovation, and provide policy support [56,65]. This, in turn, enables cities to effectively tackle various challenges and uncertainties, while mitigating the impact of economic fluctuations and bolstering the diversity and stability of the regional economy. Furthermore, fostering collaboration between the government, enterprises, and society is essential for establishing national high-tech zones and enhancing urban economic resilience. With these premises, we put forward the subsequent hypothesis 1.

**Hypothesis 1.** *The establishment of high-tech zones improve urban economic resilience in China.*

The development disparities among Chinese regions contribute to the varying effects of high-tech zones on urban economic resilience. Firstly, the eastern coastal regions of China are more advanced and possess greater economic resources and innovation advantages. Conversely, the western regions may prioritize resource-based industries and infrastructure development. These regional disparities result in diverse impacts of high-tech zones on urban economic resilience across different regions. Secondly, disparities exist in terms of industrial structure, innovation ecosystems, and policy support between cities within and outside the Yangtze River Economic Belt (YEB). The heterogeneity in the impacts of high-tech zones on urban economic resilience is further intensified by these disparities. Lastly, China's cities vary in size and development conditions, leading to differences in resource endowments. Larger cities typically possess more abundant resources and better industrial networks, thus exerting a more substantial influence on urban economic resilience through their high-tech zones. Conversely, high-tech zones in smaller cities are usually smaller in scale and less developed, leading to a comparatively smaller impact on urban economic resilience. Based on these observations, we propose hypothesis 2.

**Hypothesis 2.** *The impact of high-tech zones on urban economic resilience is spatial heterogeneity, economic belt heterogeneity, and urban size heterogeneity.*

According to Bristow and Healy [68], regions that lead in innovation have a greater ability to recover from economic crises. Their urban economic resilience depends on diversification, flexibility, and adaptability, and innovation plays a crucial role in achieving these qualities. High-tech zones create favorable conditions for urban innovation and economic growth by introducing new industries and business models [65]. This promotes the competitiveness and resilience of cities, enabling them to adapt to change, withstand shocks, and discover new growth opportunities. First, high-tech zones contribute to the development of a park economy by establishing technology-intensive collaborative R&D and innovation networks. This shift from traditional factor-dependent development to a science and technology-driven endogenous model reduces urban external dependency and enhances the city's economic resilience [69]. Second, the agglomeration of innovation resources, such as scientific research institutions, high-quality talent, financial capital, and innovative enterprises, is facilitated by high-tech zones [70]. This agglomeration enhances the urban innovation capacity and creates economies of scale, thus establishing a more competitive and diverse innovation ecosystem for the city [71,72]. Moreover, high-tech zones help build a diverse innovation ecosystem, enabling cities to effectively handle a range of opportunities and challenges, bolstering their economic resilience. Finally, high-tech zones not only drive technological innovation but also support social and policy innovation, enabling cities to tackle issues such as environmental problems and public health crises. By addressing these challenges, potential economic risks are reduced, overall efficiency is improved, and cities are better equipped to face future challenges, ultimately enhancing their sustainability and economic resilience. Taking into consideration of these observations, we put forward hypothesis 3.

**Hypothesis 3.** *High-tech zones improve urban economic resilience through a mediating path that enhances their urban innovation ability.*

Cities that have high resource allocation efficiency can significantly enhance the utilization and output efficiency of production factors [33,73,74], thereby improving their urban resilience and ability to recover from various challenges. Efficient resource allocation enables cities to optimize the utilization of human resources, capital, and technology, ensuring maximum output at minimal cost and waste. These improvements in efficiency contribute to urban economic productivity. Moreover, effective resource allocation helps in streamlining production processes and enhancing innovation capabilities, leading to the increased output efficiency of products and services. This enables cities to quickly adapt to market demand changes and enhances their potential for economic growth. Additionally, efficient resource allocation equips cities with the capability to respond to external shocks such as economic recessions, natural disasters, or other unforeseen events. Cities can flexibly adjust resource allocation, facilitating a swift resumption of normal operations and mitigating the negative impact on the economy.

High-tech zones concentrate capital, labor, and technology in technologically advanced areas while phasing out polluting, energy-intensive, and inefficient enterprises [55,64]. This shift allows for more effective resource allocation to enterprises with higher productivity, thereby fostering the emergence of new industries. Consequently, the application of production factors expands, reducing the potential economic risks in cities. Second, enterprises in high-tech zones possess unique advantages in terms of innovation and adaptation, enabling them to effectively manage costs and optimize resource allocation [75,76]. This results in lower production costs, increased productivity, and the creation of innovative products and technologies. By being adaptable to market changes and customer needs, cities can better navigate economic fluctuations and market dynamics. Finally, high-tech zones often benefit from favorable economic systems, including tax policies, financing support, intellectual property protection, and talent recruitment policies. These policies further enhance resource allocation, promote the growth of emerging industries, and strengthen urban competitiveness and resilience. Based on these observations, we propose hypothesis 4.

**Hypothesis 4.** *Optimal resource allocation plays a moderating role in the relationship between high-tech zones and urban economic resilience.*

## 3. Materials and Methods

### 3.1. Model Setting

This study investigates high-tech zones as a quasi-natural experiment. The study selects a sample of 233 cities at the prefecture-level in China from 1990 to 2021 and utilizes a difference-in-differences model to empirically examine the impact of high-tech zones on urban economic resilience. The endogeneity problem can be efficiently addressed by utilizing the difference-in-differences model, which outperforms other models in this regard. This model ensures unbiased estimates of the policy effects by effectively managing the interaction effects between the explanatory factors. By comparing the control group with the experimental group, it is possible to ascertain the net policy effect of the high-tech zones policy on urban economic resilience. The "China Torch Statistical Yearbook" is utilized to access the roster of national high-tech zones. Equation (1) presents the detailed model used in this analysis.

$$Resilience_{i,t} = \alpha_0 + \alpha_1 HTZ_{i,t} + \lambda Ctrl_{i,t} + \beta_i + \delta_t + \varepsilon_{i,t} \tag{1}$$

In this model, *i* represents the city, *t* represents time, $Resilience_{i,t}$ represents the economic resilience of city *i* in year *t*, and the explanatory variable $HTZ_{i,t}$ indicates the presence or absence of a national high-tech zone in city *i*. If a presence is detected, $HTZ_{i,t}$ takes on the value of 1, otherwise, it assumes the value of 0. $Ctrl_{i,t}$ represents the control variable, and $\beta_i$ is the city's individual fixed effect. $\delta_t$ is the time fixed effect. $\varepsilon_{i,t}$ represents the random disturbance term. The coefficient $\alpha_1$ represents the effect of national high-tech zones on urban economic resilience. A positive value for $\alpha_1$ suggests that high-tech zones can enhance urban economic resilience, while a negative value indicates a restraining effect.

### 3.2. Variable Description

#### 3.2.1. Dependent Variable

In this paper, the focal variable is urban economic resilience. We employ the measurement method of economic resilience established by Martin and Gardiner [13], which is widely accepted in the field. The urban economic resilience is determined by calculating the absolute change in the growth rate of the urban GDP. This measure provides a comprehensive insight into the economic composition of a city. As shown Equation (2),

$$Resilience_{i,t} = gdp_{i,t} \times \Delta gdpv_i \times 100 \tag{2}$$

$Resilience_{i,t}$ represents the urban economic resilience of city *i* in year *t*. $gdp_{i,t}$ indicates the standardized value of the regional GDP of city *i* in year *t*. $\Delta gdpv_i$ indicates the growth rate of $gdp_i$ in city *i* from year $t - k$ (with *k* = 1) to year *t*. The standardized value of the absolute change is multiplied by 100 for convenience in expressing the regression coefficient. Figure 2 illustrates the trends in urban economic resilience within the study area.

#### 3.2.2. Independent Variable

The high-tech zones dummy variable (HTZ) functions as the independent variable. The data were obtained from the official list of national high-tech industrial development zones that have been authorized for establishment by the State Council as of 2021. This list has been made available on the website of the Ministry of Science and Technology.

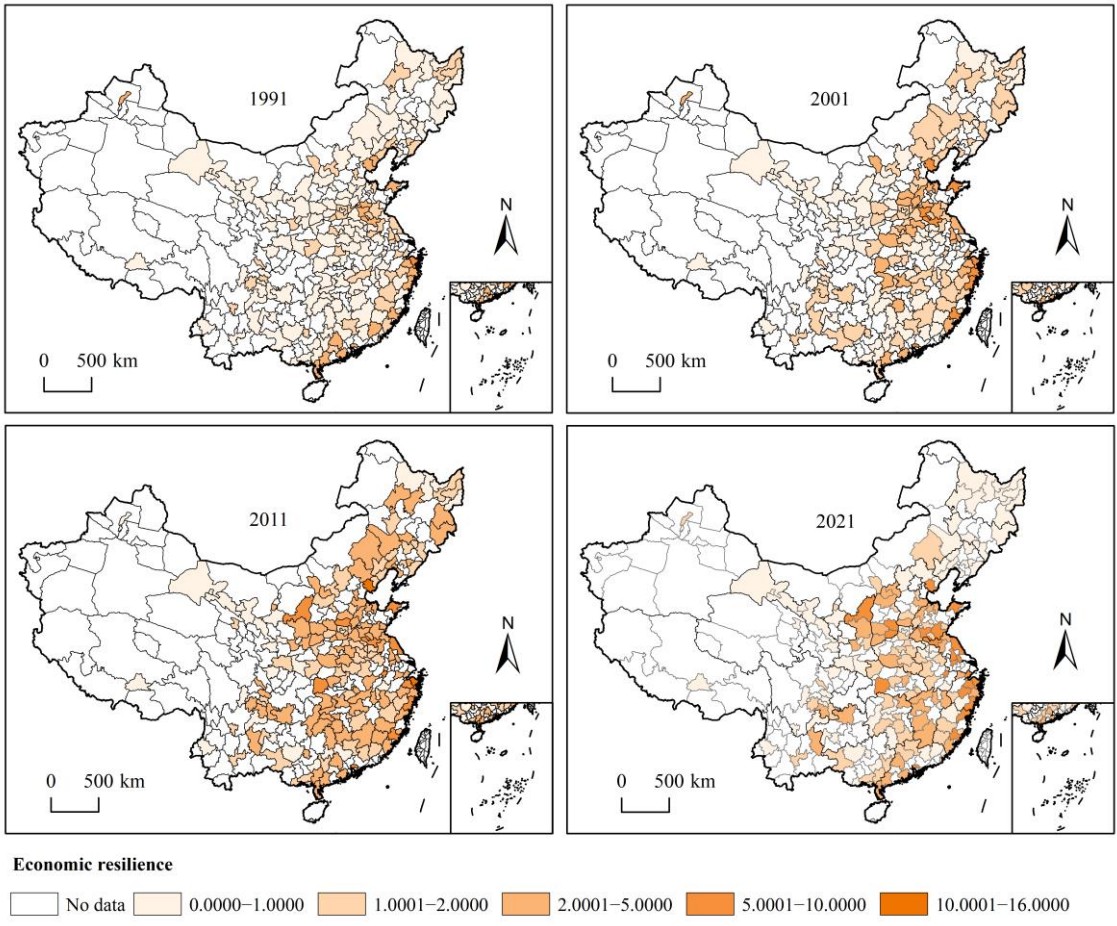

Economic resilience

No data    0.0000−1.0000    1.0001−2.0000    2.0001−5.0000    5.0001−10.0000    10.0001−16.0000

**Figure 2.** Trends in urban economic resilience in the study area.

### 3.2.3. Other Variables

Prior research has identified a range of control variables that are known to influence urban economic resilience [22,46,51,77–79], which mainly include the following: (1) Industrial structure (IS), measured by calculating the ratio of the added value of tertiary industry to the added value of secondary industry in a prefecture-level city for a specific year. (2) Financial development level (Finance), measured by calculating the logarithm of the loan balance of financial institutions in prefecture-level cities at the end of the year. (3) Educational investment level (Education), measured by taking the logarithm of the fiscal investment in education in a prefecture-level city for a given year. (4) Human capital level (Human), measured by taking the logarithm of the number of students enrolled in regular colleges and universities in prefecture-level cities for a given year. (5) Infrastructure level (Infrastructure), measured by the logarithm of the total fixed asset investment in prefecture-level cities for that year. (6) The degree of government intervention (Government), measured by taking the logarithm of the total budget spending of a prefecture-level city for the current year. (7) Degree of openness (Open), measured by the logarithm of the actual quantity of foreign capital utilized by prefecture-level cities for a given year.

### 3.3. Data

The specific information pertaining to national high-tech zones, such as their establishment dates and the cities in which they are located, was obtained from the Directory of High-tech Zones, released by the Ministry of Science and Technology of the People's Republic of China. The cities' data are collected from various reliable sources, including the China Urban Statistical Yearbook, the China Regional Economic Statistical Yearbook, the statistical yearbooks of each province, and the statistical yearbooks of each city in recent years. In order to guarantee an unbiased assessment of the influence of high-tech zones

on urban economic resilience, specific samples are eliminated. First, some prefecture-level cities (such as Lhasa) are excluded from the sample due to insufficient relevant data or because they were established as prefecture-level cities after 2010, thereby disrupting the data continuity being analyzed in this study. Second, cities that established high-tech zones in 1988, 1992, and 1993 are excluded because they are mainly municipalities and provincial capitals, which enjoy greater administrative and policy-making advantages as well as more government support and policy preferences compared with general prefecture-level cities. Including these cities would affect the results of this study. Hence, this research sample concentrates on 233 prefecture-level cities in China, spanning a time frame from 1990 to 2021. Of these, 106 cities have established national high-tech zones, serving as the experimental group. Conversely, the control group comprises the remaining 127 cities that have yet to establish high-tech zones. In order to minimize the impact of outliers, every continuous variable is downscaled by 1% and missing values are filled using linear interpolation. This process generates a well-balanced panel dataset. The statistical features of the variables in the complete sample are presented in Table 1. Table 2 presents the statistical properties of the samples taken from both the experimental group and the control group.

**Table 1.** Descriptive statistics of the full sample.

| Variable | Obs. | Mean | S.D. | Min | Max |
|---|---|---|---|---|---|
| Resilience | 7456 | 1.109 | 1.473 | 0.001 | 15.590 |
| HTZ | 7456 | 0.112 | 0.315 | 0.000 | 1.000 |
| IS | 7456 | 0.877 | 0.465 | 0.109 | 3.145 |
| Finance | 7456 | 14.870 | 1.324 | 11.670 | 19.220 |
| Education | 7456 | 11.230 | 1.749 | 6.605 | 14.780 |
| Human | 7456 | 9.064 | 1.616 | 4.691 | 12.870 |
| Infrastructure | 7456 | 14.050 | 2.171 | 9.089 | 18.280 |
| Government | 7456 | 12.880 | 1.768 | 8.698 | 16.660 |
| Open | 7456 | 8.102 | 2.285 | 2.708 | 13.470 |

**Table 2.** Descriptive statistics of the experimental and control group samples.

| Sample Group | Variable | Obs. | Mean | S.D. | Min | Max |
|---|---|---|---|---|---|---|
| experimental group | Resilience | 3392 | 1.559 | 1.848 | 0.001 | 14.460 |
| | HTZ | 3392 | 0.252 | 0.434 | 0.000 | 1.000 |
| | IS | 3392 | 0.781 | 0.350 | 0.109 | 3.145 |
| | Finance | 3392 | 15.190 | 1.342 | 11.670 | 19.220 |
| | Education | 3392 | 11.430 | 1.722 | 6.605 | 14.780 |
| | Human | 3392 | 9.599 | 1.448 | 4.691 | 12.870 |
| | Infrastructure | 3392 | 14.390 | 2.139 | 9.089 | 18.280 |
| | Government | 3392 | 13.090 | 1.726 | 8.698 | 16.660 |
| | Open | 3392 | 8.854 | 2.137 | 2.708 | 13.470 |
| control group | Resilience | 4064 | 0.752 | 0.947 | 0.001 | 15.590 |
| | HTZ | 4064 | 0.000 | 0.000 | 0.000 | 0.000 |
| | IS | 4064 | 0.952 | 0.527 | 0.109 | 3.145 |
| | Finance | 4064 | 14.620 | 1.252 | 11.670 | 18.570 |
| | Education | 4064 | 11.070 | 1.754 | 6.605 | 14.270 |
| | Human | 4064 | 8.641 | 1.617 | 4.691 | 12.600 |
| | Infrastructure | 4064 | 13.780 | 2.157 | 9.089 | 18.280 |
| | Government | 4064 | 12.710 | 1.784 | 8.698 | 15.880 |
| | Open | 4064 | 7.506 | 2.222 | 2.708 | 12.210 |

## 4. Results and Discussion

### 4.1. Empirical Analysis

The regression results of the difference-in-differences (DID) model are displayed in Table 3. The coefficients of the policy variable (HTZ) are all positive and statistically significant at the 1% level. This indicates that the establishment of high-tech zones effectively

improves urban economic resilience, providing support for hypothesis 1. By controlling for both time and city fixed effects, our findings indicate that cities with high-tech zones have an annual increase in economic resilience of 30.900% compared to those without such zones.

**Table 3.** Benchmark regression estimates.

| Variables | (1) | (2) |
|:---:|:---:|:---:|
| HTZ | 0.329 *** | 0.309 *** |
| | (5.702) | (5.397) |
| IS | | 0.0772 ** |
| | | (2.399) |
| Finance | | 0.366 *** |
| | | (12.01) |
| Education | | 0.0729 |
| | | (1.377) |
| Human | | 0.0293 |
| | | (1.437) |
| Infrastructure | | 0.0943 *** |
| | | (4.927) |
| Government | | 0.131 ** |
| | | (2.012) |
| Open | | −0.00241 |
| | | (−0.299) |
| Constant | 1.072 *** | −8.514 *** |
| | (85.84) | (−15.16) |
| City FE | YES | YES |
| Year FE | YES | YES |
| Observations | 7456 | 7456 |
| R-squared | 0.616 | 0.633 |

Note: *** and ** indicate significance at the 1% and 5% level, respectively.

High-tech zones attract enterprises, research institutions, and innovative talent to engage in technological innovation and R&D activities by setting up local innovation centers within cities [34,63]. Technological innovation and R&D activities within high-tech zones often result in spillover effects, which are shared and disseminated across cities. This promotes collaboration and knowledge exchange between different fields and industries, thereby enhancing the technical capabilities and competitiveness of other enterprises in cities. As a result, cities are better equipped to adapt to market changes and economic challenges, ultimately fostering urban economic growth and resilience. To further enhance urban economic resilience, it is of the utmost importance to optimize the development arrangement of high-tech zones, promote their high-quality development, strengthen their role as innovation engines, establish regional innovation and development hubs, and drive the stable and sustainable growth of the urban economy.

*4.2. Robustness Test*

4.2.1. Parallel Trend Test

The parallel trend assumption is an essential requirement when employing the DID approach [80]. This assumption implies that there should be no systematic difference in economic resilience change trends between cities with established high-tech zones and cities without such zones, prior to the implementation of the high-tech zones policy. In this paper, event study is used to test the parallel trend assumption [81]. The specific model is shown in Equation (3).

$$Resilience_{i,t} = \alpha_0 + \alpha_1 E_{i,t}^k + \lambda Ctrl_{i,t} + \beta_i + \delta_t + \varepsilon_{i,t} \tag{3}$$

The variables $Resilience_{i,t}$, $Ctrl_{i,t}$, $\beta_i$, $\delta_t$, and $\varepsilon_{i,t}$ are consistent with model (1). The dummy variable $E_{i,t}^k$ represents the occurrence of the 'event' of establishing a high-tech zone.

$y_i$ represents the specific year in which city $i$ establishes a high-tech zone. If $t - y_i \leq -5$, then $E_{i,t}^{-5} = 1$, otherwise $E_{i,t}^{-5} = 0$. Similarly, if $t - y_i = k$, then $E_{i,t}^{k} = 1$, otherwise $E_{i,t}^{k} = 0$ ($k \in [-5, 5]$); finally, if $t - y_i \geq 5$, then $E_{i,t}^{5+} = 1$, otherwise $E_{i,t}^{5+} = 0$.

The analysis, shown in Figure 3, presents the findings regarding this assumption. The relative duration of establishing a high-tech zone is represented on the horizontal axis, with the year of policy implementation designated as the policy base period (0). The coefficient of $E_{i,t}^{k}$ is displayed on the vertical axis. The coefficient's significance threshold at each policy time point in this figure is set at 90%.

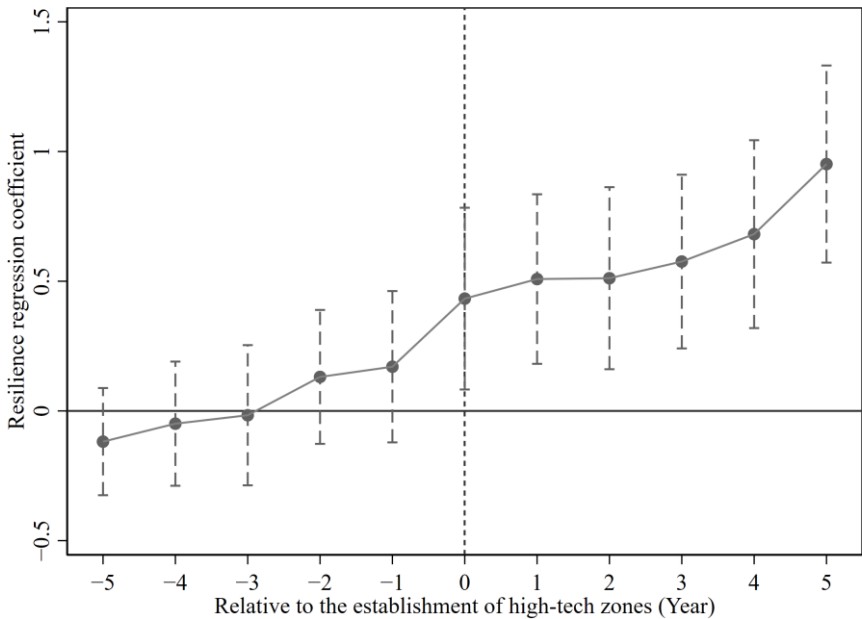

**Figure 3.** Parallel trend test results.

Based on the findings, it is evident that the regression coefficients for the period 1–5 years prior to the establishment of a high-tech zone are not statistically significant. However, they are significant for the first five years after its establishment. These outcomes imply that, prior to the implementation of the policy, there was no substantial disparity in the economic resilience levels between cities that established high-tech zones and those that did not. This offers substantiation that the benchmark model meets the assumption of a parallel trend.

4.2.2. Placebo Test

A placebo test was performed to guarantee the reliability of the benchmark regression results in this article. This test involved randomly selecting cities with established national development zones and performing 500 model estimates using the simulation process described above. Figure 4 illustrates the obtained outcomes. The diagram displays the estimated coefficient of the policy effect on the *x*-axis and the kernel density value and *p* value of the estimated coefficient on the *y*-axis. Figure 4 shows that the mean value of the estimated coefficients is 0, and a significant portion of their *p* values exceed 0.1. Furthermore, the actual estimated coefficient of the high-tech zone policy effect falls within the range of small probability events in the placebo test plot. Therefore, one can infer that the influence of high-tech zones on urban economic resilience is not haphazard. The findings of this study are robust and reliable.

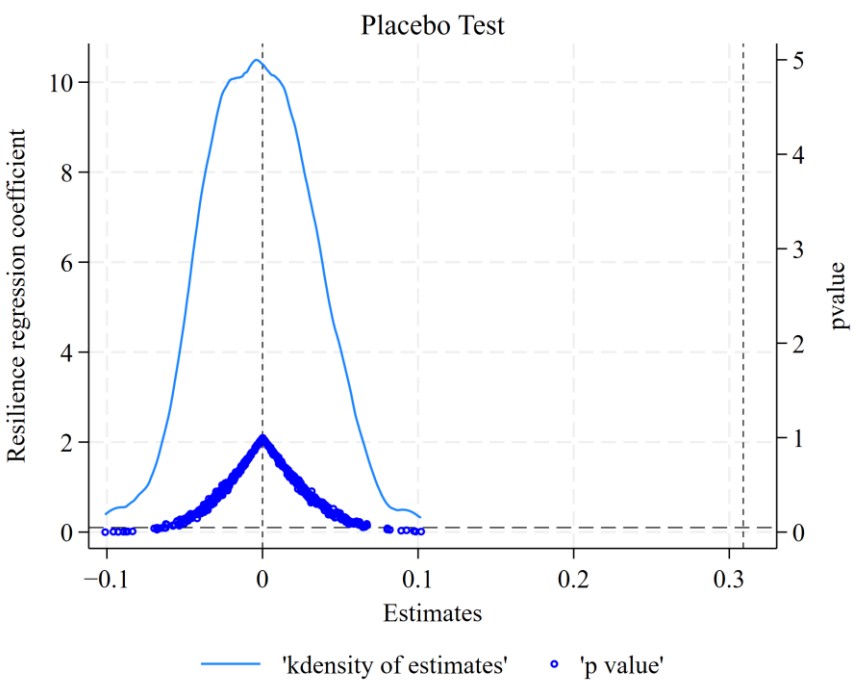

**Figure 4.** Placebo test results.

### 4.2.3. Propensity Matching Score Difference-in-Differences (PSM-DID) Method

The establishment of high-tech zones as an exogenous policy impact event has helped us to address the endogeneity problem. However, it is important to note that the selection of cities in which to establish national high-tech zones may not be random, which could introduce noise to the results of the policy evaluation. Moreover, regional differences among cities should be considered. To address the potential bias in sample selection, we utilize the PSM technique to identify a prefecture-level city in the control group that closely resembles each experimental group prefecture-level city. We then use the matched samples for model estimation. This article considers various urban characteristic variables, such as the industrial structure (IS), financial development level (Finance), educational investment level (Education), human capital level (Human), infrastructure level, the degree of government intervention (Government), and the degree of openness (Open). These variables are incorporated into a logit regression model for propensity score matching. To ensure the reliability of our matching results, we utilize three distinct methods of propensity score matching, namely K-nearest neighbor matching, radius matching, and kernel matching. Furthermore, the control group city samples are screened using the year-by-year matching method, as this article utilizes the progressive double difference method.

In Table 4, we present the findings obtained through the regression estimation model conducted using the PSM-DID method. The three columns in Table 4 illustrate the estimation results obtained from the implementation of three different PSM-DID matching methods. Regardless of the specific matching method utilized, we consistently observe a significantly positive coefficient for the policy effect at a statistical significance threshold of 1%. This discovery suggests that, even after accounting for variations in urban characteristics, the high-tech zones exert a noteworthy beneficial impact on urban economic resilience. Consequently, the PSM-DID estimation outcomes fortify the robustness of the findings articulated in this research.

### 4.2.4. Other Robustness Tests

Table 5 presents the outcomes from several other robustness tests. These tests involve manipulating the variables in different ways, such as fronting the explanatory variable, lagging the explanatory variable, substituting the explanatory variable, and adding interaction terms (HTZ × time). First, we examine the potential time lag in the establishment

of high-tech zones and its influence on the estimation results. To address the possible issue of endogeneity between the variables and HTZ, we estimated the model by fronting the dependent variable by one or two periods (Columns (1) and (2) of Table 5) and by lagging the independent variables by one or two periods (Columns (3) and (4) of Table 5), respectively. Secondly, in order to address any potential confounding factors resulting from variations in the method of measuring the dependent variable, we replaced GDP with per capita GDP when estimating the model. Column (5) in Table 5 displays the findings of the model with the replaced explanatory variable. Finally, to further strengthen the test results of the parallel trend test, we include the interaction term between the policy variable (HTZ) and the time trend variable and test based on the experimental group data. The model estimation results are shown in column (6) of Table 5. These findings demonstrate that the estimation coefficients of HTZ in all models are significantly positive, while the influence of the time trend interaction is insignificant. This suggests that our results are robust and that the establishment of national high-tech zones has a significant impact on urban economic resilience.

**Table 4.** PSM-DID estimation results.

| Variables | (1) | (2) | (3) |
|---|---|---|---|
| | K-nearest neighbor matching | radius matching | kernel matching |
| HTZ | 0.286 *** | 0.294 *** | 0.298 *** |
| | (3.316) | (5.288) | (5.362) |
| Control variables | YES | YES | YES |
| City FE | YES | YES | YES |
| Year FE | YES | YES | YES |
| Observations | 5257 | 6835 | 6838 |
| R-squared | 0.611 | 0.625 | 0.625 |

Note: *** indicates significance at the 1% level.

**Table 5.** Other robustness test results.

| Variables | (1) | (2) | (3) | (4) | (5) | (6) |
|---|---|---|---|---|---|---|
| | Future 1 | Future 2 | Lag 1 | Lag 2 | Substitution | Interaction |
| HTZ | 0.224 *** | 0.101 * | 0.224 *** | 0.106 * | 0.170 *** | 0.226 ** |
| | (3.734) | (1.658) | (3.713) | (1.731) | (2.939) | (1.962) |
| HTZ × time | | | | | | 0.010 |
| | | | | | | (0.074) |
| Control variables | YES | YES | YES | YES | YES | YES |
| Constant | −7.362 *** | −5.899 *** | −8.359 *** | −7.909 *** | −1.960 *** | −13.76 *** |
| | (−12.91) | (−10.17) | (−13.82) | (−11.91) | (−3.584) | (−12.02) |
| City FE | YES | YES | YES | YES | YES | YES |
| Year FE | YES | YES | YES | YES | YES | YES |
| Observations | 7223 | 6990 | 7223 | 6990 | 7456 | 3392 |
| R-squared | 0.630 | 0.630 | 0.633 | 0.633 | 0.486 | 0.673 |

Note: ***, **, and * indicate significance at the 1%, 5%, and 10% level, respectively.

### 4.3. Heterogeneity Analysis

#### 4.3.1. Spatial Heterogeneity

Spatial heterogeneity in cities often leads to disparities in economic development, marketization levels, and the strength of science, education, culture, and health. China's territory is divided into three regions, eastern, central, and western, based on various geographic, economic, and social factors. The implementation of the high-tech zones policy in China also reveals regional characteristics of early and pilot implementation in the eastern region, which subsequently extended to the central and western regions. Consequently, the influence of high-tech zones on urban economic resilience may vary depending on regional differences. To investigate these differences, this study conducts empirical tests on samples extracted from the three distinct city regions: the eastern, central, and western areas. The findings presented in Table 6 reveal that high-tech zones located

in the eastern region contribute 62.5% to urban economic resilience. However, the impact of high-tech zones in the central and western regions on urban economic resilience is not statistically significant. This underscores the importance of initial resource endowments in enhancing urban economic resilience.

**Table 6.** Spatial heterogeneity test results for the full sample.

| Variables | (1) | (2) | (3) |
|---|---|---|---|
| | Eastern city | Central city | Western city |
| HTZ | 0.625 *** | 0.0870 | 0.134 |
| | (5.638) | (1.229) | (1.490) |
| Control variables | YES | YES | YES |
| Constant | −13.24 *** | −4.703 *** | −7.525 *** |
| | (−11.42) | (−5.670) | (−8.529) |
| City FE | YES | YES | YES |
| Year FE | YES | YES | YES |
| Observations | 2368 | 2784 | 2304 |
| R-squared | 0.693 | 0.552 | 0.568 |

Note: *** indicates significance at the 1% level.

### 4.3.2. Economic Belt Heterogeneity

The economic development belt, known as the YEB, holds great importance in China as it stretches across the eastern, central, and western regions of the country. It encompasses the urban agglomeration located in the middle section of the Yangtze River, and Chengdu-Chongqing Urban Agglomeration [82]. Urban clusters exert a radiative influence that triggers the economic advancement of adjacent areas and assumes a pivotal role in cultivating regional harmony and urban collaboration [83]. Therefore, the impact of high-tech zones on urban economic resilience may vary depending on whether they are situated within or outside of the YEB. Thus, a comparative analysis is conducted on a sample of 90 cities situated within the YEB and 143 cities located outside the belt. According to the findings presented in Table 7, we can deduce that the urban economic resilience is influenced by the presence of high-tech zones, irrespective of their location within or outside of the YEB. Nonetheless, when considering cities within the belt, high-tech zones exhibit a much more pronounced effect on urban economic resilience than those outside the belt. This difference can be attributed to the geographical, resource, market, and policy advantages enjoyed by cities within the YEB, including their strategic location, resource abundance, diverse industrial structures, and policy support. These advantages provide greater economic potential and development opportunities, facilitating more robust economic growth through effective planning and resource management.

**Table 7.** Economic belt heterogeneity test results for the full sample.

| Variables | (1) | (2) |
|---|---|---|
| | Cities in the YEB | Cities not in the YEB |
| HTZ | 0.354 *** | 0.229 *** |
| | (3.932) | (2.983) |
| Control variables | YES | YES |
| Constant | −8.058 *** | −8.036 *** |
| | (−6.519) | (−11.79) |
| City FE | YES | YES |
| Year FE | YES | YES |
| Observations | 2880 | 4576 |
| R-squared | 0.675 | 0.611 |

Note: *** indicates significance at the 1% level.

4.3.3. Urban Size Heterogeneity

In terms of market size, diversity, resource concentration, infrastructure, and innovation and talent attraction, large cities generally have advantages. However, small and medium-sized cities have their own advantages when it comes to cost and resource management. This raises the question of whether high-tech zones in large cities versus small and medium-sized cities will have different impacts on urban economic resilience. To classify the size of cities, the State Council of China issued a Circular on Adjusting the Criteria for Classifying the Size of Cities. According to this circular, cities with a permanent urban population of more than 1 million are considered large cities, while small and medium-sized cities have permanent urban populations of less than 1 million. To conduct our analysis, we divided the sample cities into 52 large cities and 181 small and medium-sized cities. This division is based on the '2020 China Population Census Subcounty Information' compiled by the Office of the Leading Group of the Seventh National Population Census of the State Council in October 2022. Subsequently, we tested large cities and small and medium-sized cities separately.

Table 8 presents the findings from a regression analysis exploring the impact of high-tech zones on the urban economic resilience of varying sizes of cities. The findings indicate that the coefficient of HTZ is significantly positive for large cities, but not significant for small or medium-sized cities. This implies that the establishment of high-tech zones has a noticeable impact on enhancing the urban economic resilience of large cities, while it does not significantly affect smaller or medium-sized cities. One potential justification for this disparity is that high-tech zones in large cities tend to attract innovative resources, have lower transaction costs, and are more market-oriented. This facilitates positive outcomes in response to the high-tech zones policy incentives, thereby facilitating the improvement of economic resilience in large cities. Therefore, hypothesis 2 is supported.

**Table 8.** Urban size heterogeneity test results for the full sample.

| Variables | (1) | (2) |
|---|---|---|
| | Large city | Small and medium-sized cities |
| HTZ | 0.624 *** | 0.031 |
| | (4.554) | (0.641) |
| Control variables | YES | YES |
| Constant | −17.360 *** | −5.490 *** |
| | (−10.050) | (−10.700) |
| City FE | YES | YES |
| City FE | YES | YES |
| Observations | 1664 | 5792 |
| R-squared | 0.709 | 0.563 |

Note: *** indicates significance at the 1% level.

*4.4. Mechanism Test*

4.4.1. Mediating Effect of Urban Innovation Ability

According to the earlier theoretical analysis, it has been determined that the high-tech zones policy primarily enhances urban economic resilience through its main mechanism of enhancing urban innovation ability. In order to further investigate this mechanism, this paper uses a stepwise regression coefficient test based on Equations (4)–(6) [84]. First, Equation (4) is the equivalent of Equation (1). Second, we use Equation (5) to examine the relationship between HTZ and urban innovation ability (*Innovation*), with urban innovation ability serving as the dependent variable and HTZ serving as the independent variable. Finally, we construct Formula (6) by introducing economic resilience, HTZ, and urban innovation capabilities to observe any changes in the HTZ coefficient. If all three coefficients, $\alpha_1$, $\theta_1$, and $\varphi_2$, are statistically significant and the coefficient of HTZ in Formula (6) is smaller than that in Formula (4), it suggests the presence of a mediating effect.

$$Resilience_{i,t} = \alpha_0 + \alpha_1 HTZ_{i,t} + \lambda Ctrl_{i,t} + \beta_i + \delta_t + \varepsilon_{i,t} \tag{4}$$

$$Innovation_{i,t} = \theta_0 + \theta_1 HTZ_{i,t} + \lambda Ctrl_{i,t} + \beta_i + \delta_t + \varepsilon_{i,t} \tag{5}$$

$$Resilience_{i,t} = \varphi_0 + \varphi_1 HTZ_{i,t} + \varphi_2 Innovation_{i,t} + \lambda Ctrl_{i,t} + \beta_i + \delta_t + \varepsilon_{i,t} \tag{6}$$

Urban innovation ability is gauged by the score of the patents granted to the prefecture-level city in the current year. These data were obtained from the Innovation and Entrepreneurship Index compiled by Peking University's Enterprise Big Data Research Center. The test findings showing the mediating influence of urban innovation ability at the 1% significance level are shown in Table 9's columns (1) and (2). To further validate that urban innovation ability serves as an intermediary path for high-tech zones to enhance urban economic resilience, we conducted a Sobel test and Bootstrap test for its mediating effect. The test results, shown in Tables A1 and A2 in Appendix A, confirm that urban innovation capability is indeed an effective intermediary variable. Therefore, hypothesis 3 is supported.

**Table 9.** Mechanism verification results.

| Variables | Mediating Effect of Urban Innovation Ability | | Moderating Effect of Optimal Resource Allocation | |
|---|---|---|---|---|
| | (1) | (2) | (3) | (4) |
| HTZ | 1.847 *** | 0.262 *** | 0.307 *** | −0.489 *** |
| | (7.241) | (4.619) | (5.359) | (−3.552) |
| Innovation | | 0.0252 *** | | |
| | | (13.11) | | |
| Allocation | | | −0.130 | −0.194 |
| | | | (−0.848) | (−1.284) |
| HTZ* Allocation | | | | 2.254 *** |
| | | | | (5.234) |
| Control variables | YES | YES | YES | YES |
| Constant | 4.219 | −8.620 *** | −8.471 *** | −8.402 *** |
| | (1.131) | (−15.36) | (−15.01) | (−14.98) |
| City FE | YES | YES | YES | YES |
| Year FE | YES | YES | YES | YES |
| Observations | 7456 | 7456 | 7456 | 7456 |
| R-squared | 0.934 | 0.641 | 0.633 | 0.636 |

Note: *** indicates significance at the 1% level.

The results of the mediating test demonstrate that the establishment of high-tech zones has a positive impact on a city's innovation ability, thereby enhancing its economic resilience. Firstly, high-tech zones attract numerous enterprises rooted in science and technology, as well as innovative industries, diversifying the city's economy and reducing its dependence on traditional industries or a specific field. This reduces the sensitivity of the economic structure to external shocks. Secondly, high-tech zones bring together scientific research institutions and high-end talents, fostering technological innovation and industrial upgrading. This technological innovation-driven competitive advantage makes the city more resilient and adaptable to external shocks. Additionally, high-tech zones also increase the city's capacity to handle outside issues by broadening its innovation network and platform for exchanging resources. Therefore, the establishment of high-tech zones enriches the innovative potential of cities in every facet, fortifying their economic adaptability and empowering them to effectively overcome various challenges and disruptions.

### 4.4.2. Moderating Effect of Optimal Resource Allocation

Drawing on the preceding theoretical analysis, the relationship between high-tech zones and urban economic resilience is moderated by the efficient use of available resources. To delve deeper into comprehending this moderating impact, this paper construct

Equations (7) and (8). If the coefficient $\tau_3$ is statistically significant, it suggests the presence of a moderating effect.

$$Resislence_{i,t} = \omega_0 + \omega_1 HTZ_{i,t} + \omega_2 Allocation + \lambda Ctrl_{i,t} + \beta_i + \delta_t + \varepsilon_{i,t} \qquad (7)$$

$$Resislence_{i,t} = \tau_0 + +\tau_1 HTZ_{i,t} + \tau_2 Allocation + \tau_3 HTZ_{i,t} \times Allocation + \lambda Ctrl_{i,t} + \beta_i + \delta_t + \varepsilon_{i,t} \qquad (8)$$

The variable of optimal resource allocation is measured using total factor productivity (TFP), which refers to the improved efficiency in production achieved with a given level of factor inputs [85,86]. TFP is determined by the DEA-Malmquist non-parametric method. The output factor is real GDP, while the input factors are the number of employees and fixed assets (evaluated using the perpetual inventory method). The required data are obtained from the statistical yearbook of each prefecture-level city. The results of the moderating effect test are presented in Table 9, specifically in columns (3) and (4). It is evident from the results that $\tau_3$ is significantly positive. Consequently, hypothesis 4 is validated.

The results of the moderating test indicate that cities with a stronger capacity for optimizing the allocation of their urban resources can enhance the effect of their high-tech zones on urban economic resilience. This suggests that the effective allocation of urban resources, such as talents, infrastructure, finance, and industry, plays a pivotal role in enhancing the impact of high-tech zones on urban economic resilience. This is because high-tech zones typically attract a large number of talents, financial capital, industries, and other resources. Cities with a strong ability to optimize resource allocation can fully utilize these resources and channel them towards innovation activities, research projects, and industrial development, thereby strengthening the innovation leadership of high-tech zones. Consequently, the impact of high-tech zones on urban economic resilience is strengthened.

## 5. Conclusions and Policy Implications

### 5.1. Conclusions

Establishing high-tech zones aims to foster innovation, attract innovative resources, and enhance international cooperation, thus promoting the stable and sustainable development of the regional economy. Achieving sustainable development heavily relies on economic resilience, which helps cities navigate uncertainty and promote steady and sustainable economic growth in the region. Analyzing a sample of 233 cities at the prefecture-level in China from 1990 to 2021, this study investigates the empirical effects of establishing high-tech zones on urban economic resilience. The high-tech zones policy is utilized as a quasi-natural experiment for this purpose. Furthermore, this study conducts heterogeneity analysis and employs a stepwise regression coefficient test and difference-in-difference-in-difference model to test the mechanism through which the high-tech zones influence urban economic resilience. This study presents three key findings.

First, the initial findings from the benchmark regression analysis indicate that the high-tech zones policy has a positive impact on urban economic resilience. This conclusion remains valid even after implementing multiple robustness tests, such as the parallel trend test, placebo test, PSM-DID test, lag model test, and replacement of the explained variable. Second, the results based on spatial heterogeneity indicate that the establishment of national high-tech zones manifests a stronger promoting influence on the economic resilience of eastern cities. However, it does not have a promoting influence on the economic resilience of central cities and western cities. Similarly, the results derived from economic belt heterogeneity reveal that the high-tech zones policy exerts a stronger promoting effect on the economic resilience of cities in the YEB than on cities not in the YEB. Furthermore, when considering urban size heterogeneity, the high-tech zones policy significantly promotes the economic resilience of large cities, whereas it does not have any significant effect on the economic resilience of small and medium-sized cities. Finally, the mechanism test results, based on the stepwise regression coefficient test and difference-in-difference method, demonstrate that high-tech zones enhance urban economic resilience by promoting

urban innovation capabilities. Moreover, optimal resource allocation plays a moderating role in the relationship between high-tech zones and urban economic resilience.

### 5.2. Policy Implications

This study's outcomes illuminate the stabilizing function performed by high-tech zones during economic fluctuations. Additionally, these outcomes carry noteworthy implications for policymakers in China and other nations who are crafting forthcoming economic policies and strategies. First, high-tech zones serve as an effective strategic tool for enhancing the economic resilience of cities. By creating an environment conducive to innovation, industrial upgrading, and resource optimization, high-tech zones have the potential to enhance cities' ability to withstand risky shocks. This, in turn, contributes to their overall urban economic resilience. Policymakers in various countries can establish innovation incubators and science and technology parks, providing office space, laboratory facilities, and infrastructure support in order to promote stable regional growth. Additionally, financial support and tax incentives can be offered to encourage enterprises to set up or expand their operations in high-tech zones, attracting more innovative companies. Talent training programs can also be formulated to cultivate technical and managerial skills that align with the requirements of high-tech zones. By implementing talent introduction policies, the government can attract outstanding domestic and foreign talents to work and start businesses in high-tech zones, thus enhancing their competitiveness.

Second, policymakers should prioritize gaining a comprehensive understanding of urban innovation ability and optimal resource allocation when considering the impact of high-tech zones on urban economic resilience. These factors are crucial for ensuring the stability and sustainability of urban economies in the face of a changing global economy. Firstly, policymakers should focus on strengthening the innovation-leading role of high-tech zones in cities and enhancing the city's economic resilience by improving its innovation ability. Secondly, policymakers should utilize various policy tools, such as innovation incentive policies, industrial policies, and resource allocation policies, to promote innovation, facilitate industrial upgrading, optimize resource utilization, and enhance the effectiveness of high-tech zones to improve the economic resilience of cities. By adopting these approaches, cities can create more opportunities for sustainable development.

Finally, due to the significant disparities in economic scale, industrial structure, population distribution, and the comprehensive development levels among various regions of China, the Chinese government has been dedicated to reducing these regional economic development gaps through the implementation of various policies and projects, such as the 'Western Development' initiative, to foster balanced development between regions. Our research findings highlight that the impact of high-tech zones on the urban economic resilience varies significantly across different regions and sizes of city. Therefore, policymakers should tailor appropriate policies and strategies for high-tech zones to the specific conditions of each region. This approach aims to maximize the potential of high-tech zones and enable them to contribute more effectively to the stable and sustainable economic development of both western cities and small to medium-sized cities, thus promoting balanced and stable economic growth.

### 5.3. Limitations and Directions for Further Research

This paper examines the influence of high-tech zones on urban economic resilience. However, there are some limitations to this study. Future researchers should consider investigating the following two aspects: Firstly, the research findings' applicability beyond prefecture-level city areas in China could be verified by incorporating high-tech zone policies from both developing and developed countries into the sample, thereby expanding its size. Secondly, they could delve deeper into the role and mechanisms by which high-tech zones impact urban economic resilience. Given the complex and varied impacts of high-tech zones on urban economic resilience, it is crucial to explore them from various

perspectives to enhance government guidance and maximize the benefits of high-tech zones in terms of urban economic resilience.

**Author Contributions:** Conceptualization, R.Y. and X.X.; methodology, X.X.; software, R.Y.; validation, T.H., S.Z. and W.Z.; formal analysis, R.Y.; investigation, R.Y.; resources, X.X.; data curation, R.Y.; writing—original draft preparation, R.Y. and X.X.; writing—review and editing, W.Z.; visualization, R.Y.; supervision, S.Z.; project administration, S.Z. All authors have read and agreed to the published version of the manuscript.

**Funding:** This research received no external funding.

**Data Availability Statement:** Data is unavailable due to privacy.

**Conflicts of Interest:** The authors declare no conflict of interest.

## Appendix A

**Table A1.** Sobel test results.

|  | **Coef** | **Std Err** | **Z** | **P > |Z|** |
|---|---|---|---|---|
| Sobel | 0.047 | 0.008 | 5.971 | $2.364 \times 10^{-9}$ |
| Goodman-1 (Aroian) | 0.047 | 0.008 | 5.957 | $2.577 \times 10^{-9}$ |
| Goodman-2 | 0.047 | 0.008 | 5.985 | $2.167 \times 10^{-9}$ |
| a coefficient | 1.847 | 0.274 | 6.738 | $1.600 \times 10^{-11}$ |
| b coefficient | 0.025 | 0.002 | 12.881 | 0.000 |
| Indirect effect | 0.047 | 0.008 | 5.971 | $2.400 \times 10^{-9}$ |
| Direct effect | 0.262 | 0.046 | 5.751 | $8.900 \times 10^{-9}$ |
| Total effect | 0.309 | 0.046 | 6.716 | $1.900 \times 10^{-11}$ |
| Proportion of total effect that is mediated |  |  | 0.151 |  |
| Ratio of indirect to direct effect |  |  | 0.178 |  |
| Ratio of total to direct effect |  |  | 1.178 |  |

**Table A2.** Bootstrap test results.

|  | **Observed Coefficient** | **Bootstrap Std. Err.** | **Z** | **P > |Z|** | **Normal-Based [95% Conf. Interval]** | |
|---|---|---|---|---|---|---|
| Indirect effect | 0.047 | 0.007 | 6.330 | 0.000 | 0.032 | 0.061 |
| Direct effect | 0.262 | 0.055 | 4.750 | 0.000 | 0.154 | 0.370 |

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
