# Peer review of "Has the Establishment of High-Tech Zones Improved Urban Economic Resilience? Evidence from Prefecture-Level Cities in China"

_land, doi:10.3390/land13020241_

Round 1
Reviewer 1 Report
Comments and Suggestions for Authors
I have now completed the review of your paper entitled "Has the Establishment of High-Tech Zones Improved Urban Economic Resilience? Evidence from Prefecture-Level Cities in China". It aims to assess the impact of high-tech zones on urban economic resilience in China using a difference-in-differences approach. The authors have made a good attempt at evaluating this policy tool. The results are meaningful and reveal several noteworthy insights. The methods are strong and utilize appropriate tests. However, there are concerns regarding the breadth of the scope, rigor surrounding key assumptions, and specificity of policy implications that require addressing.
- The introduction provides good background but could be more concise. Consider shortening the policy background section to 1-2 paragraphs focusing only on key information needed to frame the study.
- Hypothesis 1 seems quite broad. Consider narrowing the scope by focusing on a specific dimension of economic resilience that high-tech zones are expected to influence.
- The scope literature review needs to be broadened in order to contextualize this study in the existing scholarly debate. The following relevant studies are suggested:
https://www.mdpi.com/2073-445X/12/2/283
https://doi.org/10.3389/fsufs.2023.1191024
- In the data section, clarify how missing values were addressed. Were they simply interpolated or were statistical methods used to estimate them? This could bias results.
- The parallel trends assumption is assessed only visually. Consider augmenting this with a statistical test by interacting the policy variable with a time trend.
- In the heterogeneity analysis, also consider the moderating effects of high-tech zone characteristics, such as size, specialization, incentives offered, etc. This may explain variation.
- The proposed mediating role of innovation capability seems reasonable. However, the evidence presented relies only on a Baron and Kenny approach. Use more rigorous mediation analysis methods.
- There are a few vague statements in the conclusion related to policy implications (e.g. "cultivate more competitive high-tech zones"). Consider removing or replacing these with specific, actionable suggestions.
- While well written overall, there are areas, especially surrounding the proposed mechanisms, where discussion could be expanded and clarified. Carefully review these sections.
Reviewer 2 Report
Comments and Suggestions for Authors
The comment is attached

Reviewer 3 Report
Comments and Suggestions for Authors
This paper discusses the establishment of high-tech zones in China as a strategic policy tool to foster urban scientific and technological innovation and ensure stable economic growth. The study utilizes 233 prefecture-level cities in China from 1990 to 2021 as research samples, employing a difference-in-differences model to empirically analyze the impact of high-tech zones on urban economic resilience. The findings indicate a significant positive effect of high-tech zones on economic resilience, with variations across regions and city sizes. The study highlights the importance of enhancing innovation capabilities and optimizing resource allocation as key mechanisms through which high-tech zones influence urban economic resilience. Overall, the paper contributes to the existing literature by providing empirical evidence on the regional-level impact of high-tech zone policies on economic resilience.
This paper explores the objective of establishing high-tech zones in China to promote stable and sustainable regional economic development by fostering innovation, attracting innovative resources, and strengthening international cooperation. Using a sample of 233 prefecture-level cities from 1990 to 2021, the study employs the national high-tech zone policy as a quasi-natural experiment to analyze its impact on urban economic resilience. The research concludes that the policy has a positive effect on economic resilience, with variations across regions, city sizes, and economic belts. The study suggests that high-tech zones enhance urban economic resilience by promoting innovation capabilities and optimizing resource allocation. The findings emphasize the stabilizing role of high-tech zones in economic fluctuations and provide policy implications for policymakers, recommending the prioritization of innovation capacity and resource allocation strategies to improve economic resilience. The text also underscores the need for tailored policies based on regional conditions to maximize the potential contribution of high-tech zones to balanced and stable economic growth.
In conlusion, I agree with the publication of this paper.
Reviewer 4 Report
Comments and Suggestions for Authors
The paper is well written, properly organized and of high scientific soundness and I recommend it to be published.
I would, however, suggest to improve it by adding some missing information about:
1. the way innovation is measured when testing the mediating effect (what is its indicator?, what is the source of it?)
2. the way allocation is measured when testing the moderating effect (the authors mention that TFP derived from DEA usage is adopted here; however, in DEA some multiple inputs and outputs are used to get the efficiency score – What are they? What are their indicators? What is the source of the data?)
Moreover, some arguments for using DID method would be expected – what are the advantages of the method justifying its choice?
In Conclusion, I would also expect some generalization of the results – is it possible to use the Chinese experience with high-tech zones in any other economy? Are the results in any way can be generalized to form conclusions about such kind of support for economic resilience of a city (in any place of the world)? What are the limitations of the results?
Round 2
Reviewer 1 Report
Comments and Suggestions for Authors
The authors have satisfactorily addressed the majority of issues raised in the previous review. The policy background is more concise and focused, the hypothesis scope narrowed appropriately. The treatment of missing data and assumptions is now clearly explained.
However, a few areas still need refinement:
- The statistical test for parallel trends relies on a simple interaction. Consider using a more robust falsification test like an event study to assess pre-trends.
- Literature review still needs to be enhanced in the light of previous comments.
- The discussion surrounding high-tech zone characteristics is still vague. Be more specific about what traits were examined and how they moderate effects.
- The mechanisms section, while improved, could still use some tightening of explanations and smoothing of logic flow. Carefully review.
- The policy implications emphasize general innovation capacity and resources. Relate recommendations more tightly to the high-tech zone context based on your findings.
Reviewer 2 Report
Comments and Suggestions for Authors
I appreciate the authors' effort. Now the paper is written clearly and therefore is more understandable and interesting.
